**Subject Category:**
Biology (whole organism)

ecology/developmental biology

calliphoridae, forensic entomology,
developmental model, life history

**Authors for correspondence:**
Jens Amendt
e-mail: amendt@em.uni-frankfurt.de
Jiangfeng Wang
e-mail: jfwang@suda.edu.cn

†The first two authors should be considered as joint first author.

# Temperature-dependent development of the blow fly *Chrysomya pinguis* and its significance in estimating postmortem interval

Yingna Zhang[1,2,†], Yu Wang[1,†], Jian Sun[3], Guoliang Hu[1],
Man Wang[1], Jens Amendt[4] and Jiangfeng Wang[1]

[1]Department of Forensic Medicine, Soochow University, Ganjiang East Road, Suzhou 215000, People's Republic of China
[2]Department of Anatomy, Shihezi University, Beisi Road, Shihezi, People's Republic of China
[3]Criminal Police Branch, Wuxi Public Security Bureau, Qianhu Road, Wuxi, People's Republic of China
[4]Institute of Forensic Medicine, University of Frankfurt, Kennedyallee, Frankfurt/Main, Germany

 YZ, 0000-0002-8746-5788; JW, 0000-0001-7642-6727

*Chrysomya pinguis* (Walker) (Diptera: Calliphoridae) is an endemic Asiatic blow fly species of forensic importance. *Chrysomya pinguis* is one of the first species to colonize a corpse, especially in high altitude areas during spring and autumn when the ambient temperature is lower. Despite its potential for forensic investigations to estimate the minimum postmortem interval (PMI_min), little is known about the development of *C. pinguis*. In this study, *C. pinguis* was collected from the Yangtze River Delta region of China and reared at seven constant temperatures between 16°C and 34°C to investigate the effect of temperature on development duration, accumulated degree hours and larval body length of *C. pinguis*. Isomorphen and isomegalen diagrams for *C. pinguis* were generated using the results, and equations describing the variation in larval body length during development and the temperature-induced variation in development time were also obtained. *Chrysomya pinguis* can complete its life cycle at 16–34°C. The mean (±s.d.) developmental durations of *C. pinguis* from egg to adult at 16°C, 19°C, 22°C, 25°C, 28°C, 31°C and 34°C were 811.0 ± 3.8, 544.8 ± 2.0, 379.8 ± 1.8, 306.7 ± 2.4, 250.0 ± 2.8, 203.2 ± 2.1 and 185.3 ± 1.6 h, respectively. The mean (±s.e.) developmental threshold temperature D_0 and the thermal summation constant K of the whole developmental process of *C. pinguis*

were estimated as $10.88 \pm 0.21°C$ and $4256.50 \pm 104.50$ degree hours, respectively. This study provides fundamental development data for the use of *C. pinguis* to estimate $PMI_{min}$.

## 1. Introduction

Calliphorids, commonly known as blow flies, play an important role in the decomposition process of animals in most terrestrial ecosystems [1]. As they are often the first to arrive and oviposit on a dead body, the oldest individual of Calliphorids collected on the corpse is often used to obtain a minimum postmortem interval ($PMI_{min}$) for solving cases involving death [2–5]. As blow flies are ectotherms and their developmental time is species specific [6,7], it is important to accurately identify the blow fly species and reconstruct temperature profiles of the scene to correctly estimate the $PMI_{min}$.

Entomological methods to age the immature insects are generally achieved through identifying developmental landmarks, such as length, weight and developmental stages of the juvenile insect, depending on the ambient temperature [8]. To estimate $PMI_{min}$ more easily, several developmental models have been proposed, and the thermal summation model and the isomorphen as well as the isomegalen diagrams have been the most commonly applied models in forensic entomology [9].

*Chrysomya pinguis* (Walker) (Diptera: Calliphoridae) is an important necrophagous species that is primarily distributed in Asian countries [10], such as China (figure 1), Japan, Korea, The Philippines, Vietnam, Laos, Thailand, Malaysia, Indonesia, Nepal, India, Sri Lanka and Pakistan [11]. Insect succession studies have shown that *C. pinguis* often coexists with *Chrysomya megacephala* (Fabricius) [12], but *C. pinguis* appears to be adapted to colder climates [13] and is more commonly found in forested and highland areas [14,15]. Some entomologists have suggested that *C. pinguis* migrates between low and high elevations depending on the season [16]. It is well established that *C. pinguis* colonizes human corpses in different areas of Asia, and is regarded as a forensically important species [10,17].

At present, the majority of studies on *C. pinguis* have been focused on its external morphology [10,18,19], submicroscopic structure [20–22], karyotypes [23], geographical abundance [24–27], myasis [28] and DNA-based techniques used for species identification [17,29,30]. However, there have not been any developmental studies of this species [31]. Therefore, we studied larval growth, development duration and thermal summation of *C. pinguis* under seven constant temperatures between 16–34°C. Using these data, developmental models were built to improve the use of this species for forensic investigations to estimate the $PMI_{min}$.

## 2. Material and methods

### 2.1. Establishment of laboratory colony

In April 2016, about 20 fly adults were collected from pig cadavers put in a field of Suzhou, China (31°21′ N, 120°53′ E). We obtained the permissions to carry out fieldwork by Forensic Autopsy Centre of Suzhou. Among the flies, eight adults (female : male = 3 : 1) had very similar morphology as *C. megacephala*. Flies with brilliant blue body colour and grey bucca with black hairs were picked out and caged (30 × 30 × 30 cm³) at room temperature (approx. 25°C) with natural humidity and light at the laboratory of Forensic Entomology, Suzhou. Adults were fed with fresh water and a 1 : 1 mixture of sugar and powdered milk. Fresh lean pork was provided to induce ovary development and reproduction. Eggs that were oviposited on the fresh lean pork were removed from the cage and reared in a separate insect-rearing box under controlled conditions until emergence. Newly emerged adults from each rearing box were caged separately. Adults were randomly selected from each cage and pinned specimens were made, all of which were subsequently identified as *C. pinguis* under stereomicroscope using external morphology and male genitalia based on Fan [32]. *Chrysomya pinguis* is a large-sized species with high flight capability, therefore flies were combined into one larger cage (200 × 100 × 100 cm³) to provide more space for mating and oviposition. In May 2017, about 300 wandering larvae were collected from the same field environment and cultured as above. After eclosion, the adults identified as *C. pinguis* were supplemented into the established colony. The colony was cultured for three generations and maintained at a density of approximately 4000 adults per cage for subsequent experiments.

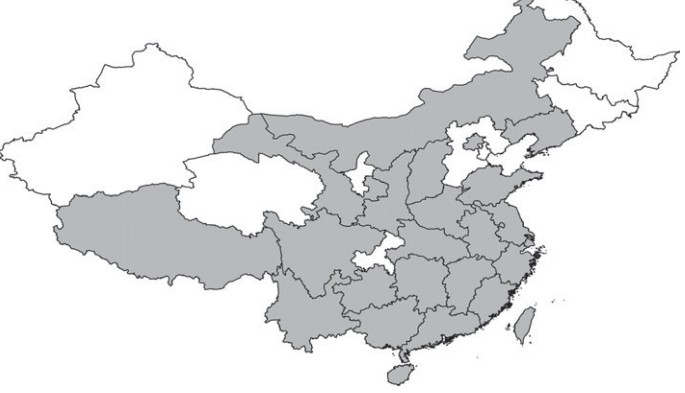

**Figure 1.** Grey areas denote the distribution of *C. pinguis* in China.

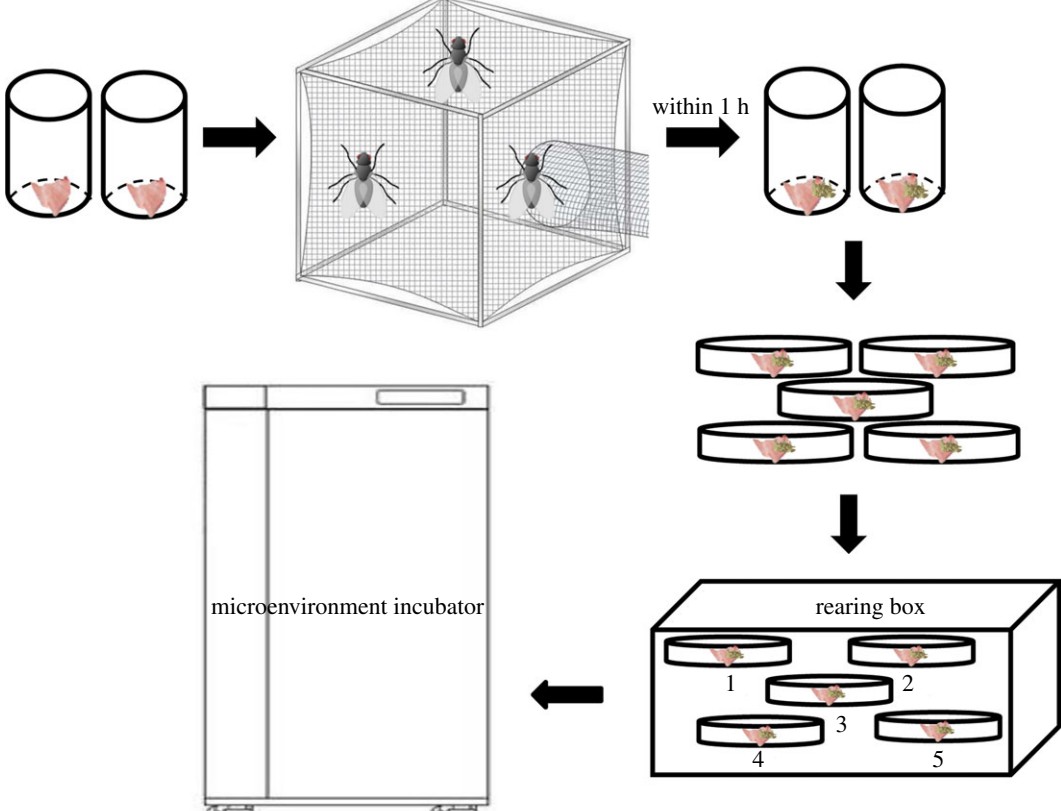

**Figure 2.** The procedure of eggs collection and placement.

## 2.2. Development duration and larval body length

Two plastic bottles of 10 cm in height and 8 cm in diameter were used to place 10 g fresh lean pork, and the bottles were placed in the insect-rearing cage to induce oviposition. Eggs laid within 1 h were collected from the bottles. About 1000 eggs, counted as one replicate, were separated into five parts and placed separately in five 10 cm Petri dishes. After 20 g fresh lean pork were added to each Petri dish, all dishes were numbered and moved into an insect-rearing box ($32 \times 22 \times 10$ cm$^3$) that contained sandy soil at the bottom. The insect-rearing box was kept in the artificial climate incubator LHP-300H (Yingmin Co. Ltd, Suzhou, China) at constant temperatures of 16°C, 19°C, 22°C, 25°C, 28°C, 31°C or 34°C, with 75% humidity and photoperiod of 12 L : 12 D (figure 2).

The culture dishes were checked every hour to monitor the hatching of eggs. Once larvae began to hatch, meat slices about 5 mm thick were placed in the Petri dish to distribute larvae more evenly within the Petri dish based on the larvae's food consumption. One of the five dishes was chosen

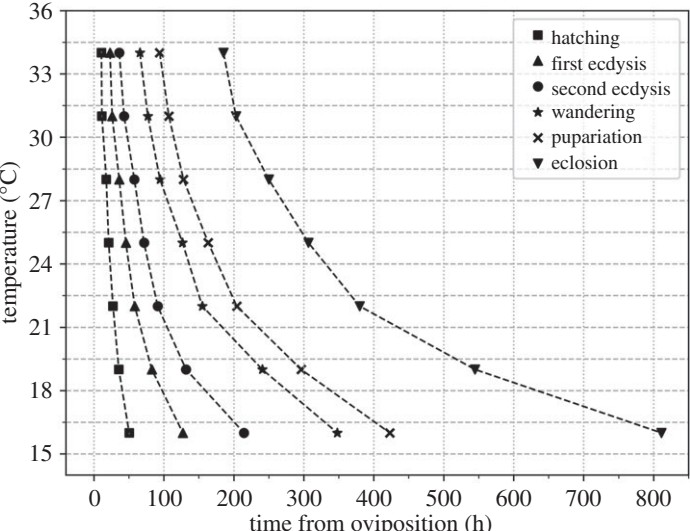

**Figure 3.** Isomorphen diagram of *C. pinguis*. The duration of each developmental milestone (hatching, first ecdysis, second ecdysis, wandering, pupariation and eclosion) was plotted with the time from oviposition to the onset of each milestone. Each curve corresponds to a developmental milestone.

randomly in an order determined using a random number generator, and eight larvae were selected from the dish until larvae were in the wandering stage. Samples were killed in hot water (at least 90°C) for 30 s and stored in 80% ethanol [33]. Samples were generally measured within a week of sampling. The larval length was measured by a digital vernier caliper with a precision of 0.01 mm (Shengong, Shanghai). The larval instar was determined using a calibrated stereomicroscope (Carl Zeiss, Göttingen, Germany). In the wandering and intra-puparial stages, observations were conducted every 4 h until the first adult emerged. The above experiment was repeated five times for each temperature in different incubators.

## 2.3. Data analysis

Data analysis was performed using Python 3.6.5 and R 3.5.1. In order to develop an equation to estimate the $PMI_{min}$, the relationship between larval body length and time after hatching was investigated through a polynomial regression analysis adopting 'larval body length' as an independent variable and 'the time after hatching' as a dependent variable. The equation was as follows: $L = aT^3 + bT^2 + cT + d$, where $L$ = larval body length (mm), $T$ = the time after hatching (d) and $a$, $b$, $c$, $d$ are constants. As forensic investigators commonly use larval body length to determine developmental time, we also conducted a polynomial regression analysis adopting 'the time after hatching' as the independent variable and 'larval body length' as the dependent variable [34], and the formula was: $T = aL^3 + bL^2 + cL + d$, where $T$ = the time after hatching (d), $L$ = larval body length (mm) and $a$, $b$, $c$, $d$ are constants.

The regression model revised by Ikemoto & Takai [35] was used to detail the relationship between developmental time and accumulated degree hour (ADH) of each developmental stage, as well as the whole developmental process. The formula is as follows: $y = K + D_0 x$, where $D_0$ and $K$ are constants, and $D_0$ = developmental threshold temperature and $K$ = thermal summation constant.

# 3. Results

## 3.1. Developmental duration of each stage and isomorphen diagram

The whole development process of *C. pinguis* from egg to adult can be completed between 16°C and 34°C. The raw data of life stages related to sample ages were shown in electronic supplementary material, table S1, and the portions of individuals observed in different life stages were given. The data on development time were used to generate isomorphen diagrams of duration by different developmental events at different constant temperatures (figure 3).

Next, thermal summation models were established through linear regression analyses of the relationship between ADH, the duration of six developmental stages and the total developmental

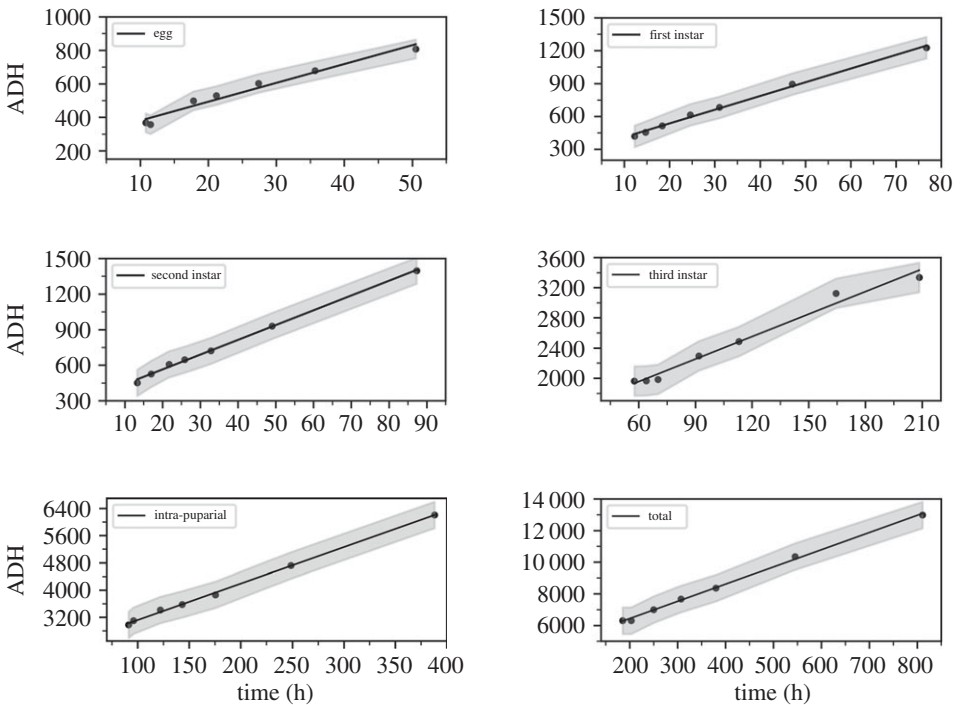

**Figure 4.** Thermal summation models of six developmental stages and total development duration of *C. pinguis*. Grey shadow represents the 95% confidence interval.

**Table 1.** Developmental threshold temperatures ($D_0$) and thermal summation constants (K) for six developmental stages and the total development period of *C. pinguis*, and the coefficient of determination ($R^2$) of thermal summation models, calculated using the method described by Ikemoto & Takai [35] (s.e. = standard error).

| developmental stages | K ± s.e. (degree hours) | $D_0$ ± s.e. (°C) | $R^2$ |
|---|---|---|---|
| egg | 267.62 ± 26.08 | 11.25 ± 0.92 | 0.97 |
| first instar | 285.14 ± 13.39 | 12.51 ± 0.35 | 0.99 |
| second instar | 315.15 ± 11.57 | 12.46 ± 0.27 | 0.99 |
| third instar | 1357.20 ± 72.06 | 9.94 ± 0.59 | 0.98 |
| hatching-pupariation | 1931.10 ± 65.71 | 11.06 ± 0.32 | 0.99 |
| intra-puparial | 2043.40 ± 37.01 | 10.72 ± 0.18 | 0.99 |
| total duration | 4256.50 ± 96.79 | 10.88 ± 0.22 | 0.99 |

process (figure 4). Using the thermal summation model, developmental threshold temperatures ($D_0$) and thermal summation constants (K) of *C. pinguis* were calculated for each developmental stage and the whole developmental process (table 1). With the exception of the regression model of the egg stage, the coefficient of determination ($R^2$) of the other stages was above 0.98, indicating that the fit of the model was high. The developmental threshold temperature ($D_0$) and the thermal summation constant (K) of the duration from first ecdysis to pupariation were 11.06°C and 1931.10 degree hours, and $D_0$ and K of the whole developmental process of *C. pinguis* were 10.88°C and 4256.50 degree hours, respectively.

## 3.2. Changes in larval body length and establishment of isomegalen diagram

The variation curve of larval body length from hatching to the wandering stage is shown in figure 5. Regression analysis was carried out with 'time after hatching' as the independent variable and 'larval body length' as the dependent variable to obtain the equation describing the change in larval body length with time under different constant temperatures (table 2). Another regression analysis was

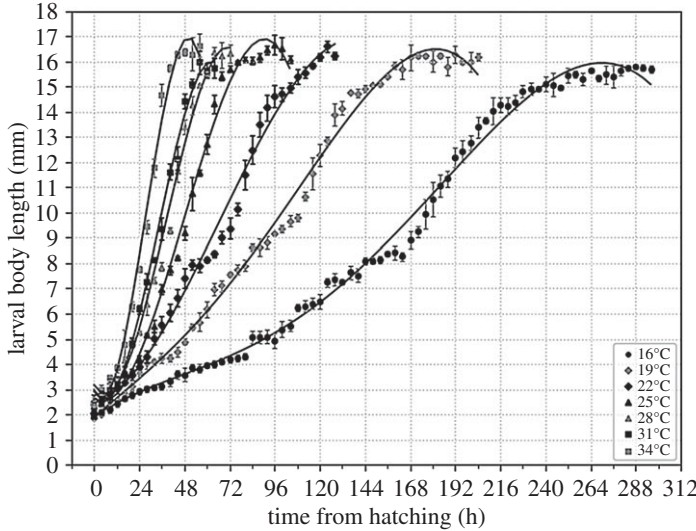

**Figure 5.** Curves of larval body length of *C. pinguis* variation with time at different temperatures. The vertical bars represent the standard deviation. Each point represents the average length of 8*5 = 40 larvae.

**Table 2.** Simulation equations, degrees of freedom (d.f.) and coefficient of determination ($R^2$) of the relationship between the body length of *C. pinguis* larvae (*L*) (mm) and the time after hatching (*T*) (d) at seven constant temperatures ($p < 0.001$).

| temperature (°C) | equation | d.f. | $R^2$ |
|---|---|---|---|
| 16 | $L = -0.0172T^3 + 0.3405T^2 - 0.5088T + 2.90$ | 75 | 0.988 |
| 19 | $L = -0.0545T^3 + 0.6256T^2 + 0.2196T + 2.40$ | 52 | 0.992 |
| 22 | $L = -0.1414T^3 + 1.1793T^2 + 0.3973T + 2.47$ | 33 | 0.991 |
| 25 | $L = -0.6269T^3 + 3.7363T^2 - 1.399T + 2.64$ | 27 | 0.994 |
| 28 | $L = -1.3953T^3 + 6.6932T^2 - 3.0622T + 3.17$ | 19 | 0.991 |
| 31 | $L = -1.965T^3 + 8.3341T^2 - 3.3489T + 2.91$ | 17 | 0.995 |
| 34 | $L = -4.0575T^3 + 13.8267T^2 - 4.4897T + 2.98$ | 15 | 0.992 |

**Table 3.** Simulation equations, degrees of freedom (d.f.) and coefficient of determination ($R^2$) of the relationship between the time after hatching (*T*) (d) and the body length of *C. pinguis* larvae (*L*) (mm) at seven constant temperatures ($p < 0.001$).

| temperature (°C) | equation | d.f. | $R^2$ |
|---|---|---|---|
| 16 | $T = 0.0085L^3 - 0.2476L^2 + 2.8293L - 5.18$ | 75 | 0.989 |
| 19 | $T = 0.0041L^3 - 0.1124L^2 + 1.3526L - 2.39$ | 52 | 0.978 |
| 22 | $T = 0.0024L^3 - 0.0687L^2 + 0.908L - 1.69$ | 33 | 0.992 |
| 25 | $T = 0.0035L^3 - 0.0984L^2 + 1.0155L - 1.83$ | 27 | 0.971 |
| 28 | $T = 0.0025L^3 - 0.0743L^2 + 0.8235L - 1.64$ | 19 | 0.993 |
| 31 | $T = 0.0019L^3 - 0.0563L^2 + 0.6353L - 1.14$ | 17 | 0.983 |
| 34 | $T = 0.0023L^3 - 0.0676L^2 + 0.6898L - 1.33$ | 15 | 0.979 |

conducted with 'larval body length' as the independent variable and 'time after hatching' as the dependent variable to obtain the equation describing the differences in time after hatching with larval body length under different constant temperatures (table 3). The equations provided above are both descriptive equations, which reflected the data from the present study well. While the equation

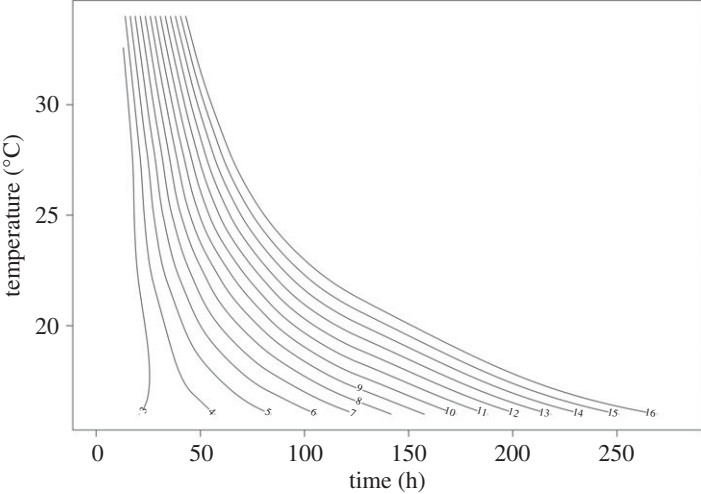

**Figure 6.** Isomegalen diagram of *C. pinguis*. Each contour represents larval body length (*Z*-axis) in relation to their age (*X*-axis) and the temperature the larvae were exposed to during development (*Y*-axis). The dimension of the *Z*-axis (length) is not needed visually to calculate a PMI$_{min}$, and is therefore, not illustrated in the figure.

parameters do not have biological significance, they allow for the model to fit better. The isomegalen diagram was established using the larval body length data (figure 6). This development model allows for the calculation of the age of larvae using body length before the post-feeding stage.

# 4. Discussion

Many studies have highlighted the forensic value of *C. pinguis*, but the data on its development had not yet been reported. This study provides the basic developmental data of *C. pinguis* and development models including thermal summation model, and isomorphen and isomegalen diagrams for a range of seven temperatures between 16°C and 34°C. These developmental models facilitate the estimation of PMI$_{min}$. Sang *et al*. [36] reported that at a crime scene with an average temperature of 21.76°C, the oldest larvae of *C. pinguis* collected from corpse were in the third instar stage. Larval body length was not mentioned but the specimens had an emptied crop, which indicated that the larvae had entered the wandering stage. Based on this information, our isomorphen diagram estimates that the time required for *C. pinguis* from egg development to third instar at 21.76°C would be 6.8 days and the time from egg development to pupariation would be 8.9 days, indicating that the age of the oldest larvae in the wandering stage collected from the scene or the PMI$_{min}$ was 6.8–8.9 days. The investigators identified that the person had been decreased 10 days before the discovery of its corpse. Our PMI$_{min}$ estimate based on the basic morphological data presented in this past study suggests that our model is capable of identifying relatively good estimates of PMI$_{min}$.

In our study, *C. pinguis* was able to complete development at 16°C to 34°C, which is consistent with a laboratory study of temperature adaptation in *C. pinguis* by Yang & Shiao [12], who found that *C. pinguis* is able to complete its development at 15–35°C. However, at 38°C, *C. pinguis* did not complete development, indicating that the maximum resistance temperature of this species is between 35°C and 38°C. Our field studies have shown that *C. pinguis* is a dominant species in cooler seasons, and similar findings have been obtained in other studies. Kano *et al*. [13] and Tachibana & Numata [16] showed that *C. pinguis* has a preference for areas and seasons with low temperature, and its peak population is always in spring and autumn. In Japan, *C. pinguis* adults have been found in mountainous areas during the summer and they may migrate from the highlands to the lowlands in spring and autumn as the temperature cools down [16]. Therefore *C. pinguis* may be of high value for estimating PMI$_{min}$ in cooler environments, or may be used to determine whether a corpse was exposed during the summer according to the occurrence of this species.

In forensic entomology development study, in order to record the body length and/or instar of a species, their larvae were usually reared at different temperatures, and individuals were removed periodically (usually without replacement) [37–40]. There are usually two sampling strategies. One is to select a narrower sampling interval at higher temperatures and a wider sampling interval at lower

temperatures. The other is to select fixed sampling interval, just like the sampling method we used in the study. Though the method we used is more convenient as we do not need to change our sampling interval at different temperatures, it usually consumes more samples, especially at lower temperature. For example, over half of the larvae (more than 500) were sampled from the original cohort at 16°C in this study, which may have a potential effect on the development of *C. pinguis*. An evaluation of sampling methods study by Wells *et al.* [41] revealed that random subsamples would produce a predictive model similar to that of the full dataset, while subsampling the largest larvae produced a predictive model that performed poorly. The larvae were randomly sampled in this study, which means that our sampling method may have little effect on the development. Further studies are needed to verify this.

Recent studies have shown that geographically distinct populations can vary in developmental time [5,38,42]. Development time differences have been found in *Calliphora vicina* Robineau-Desvoidy, 1830 and *Lucilia sericata* (Meigen, 1826) (Diptera, Calliphoridae) from geographically separate populations [43–45]. For example, at 33.5°C, *L. sericata* from the Michigan strain developed fastest, whereas *L. sericata* from the California strain took the longest to mature. Differences between the findings for the same species could be a result of geographically separate populations. Therefore, it is important to collect precise development data in different geographical regions of the world so as to provide developmental data of the species in or around the particular crime scene when estimating the PMI$_{min}$ [5,46,47].

Ethics. Animal care approval was received from the Ethics Committee of Soochow University (ECSU-20190000109). The fieldworks were permitted by the Forensic Autopsy Centre of Suzhou.

Data accessibility. Data available from the Dryad Digital Repository at: https://doi.org/10.5061/dryad.n0pf381 [48].

Authors' contributions. Y.Z. carried out the laboratory work, participated in data analysis, participated in the design of the study and drafted the manuscript; Y.W. collected field data and critically revised the manuscript; G.H. participated in data analysis and revised the manuscript; M.W. and J.S. coordinated the study; J.A. and J.W. conceived of the study, designed the study and helped draft the manuscript. All authors gave final approval for publication.

Competing interests. We have no competing interests.

Funding. This work was supported by the National Natural Science Foundation of China (grant no. 31872258), China Postdoctoral Science Foundation (grant no. 2017M621819) and Priority Academic Program Development of Jiangsu Higher Education.

Acknowledgements. We thank Forensic Autopsy Centre of Suzhou offering unrestricted field and laboratory access.

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
