## [Reviewer comments · Royal Society Open Science]

Review History

RSOS-190003.R0 (Original submission)

Review form: Reviewer 1 (Jeffrey Wells)

Is the manuscript scientifically sound in its present form?

Yes

Are the interpretations and conclusions justified by the results?

No

Is the language acceptable?

Yes

Is it clear how to access all supporting data?

Yes

Do you have any ethical concerns with this paper?

No

Have you any concerns about statistical analyses in this paper?

No

Recommendation?

Major revision is needed (please make suggestions in comments)

Comments to the Author(s)

Chrysomya pinguis is a widespread and common blow fly. Published development data for *C. pinguis* are not available and could be very useful to death investigators. The manuscript is generally well written, and it potentially merits publication if the authors can revise the methods and data presentation.

MAJOR COMMENTS

-The sampling scheme is not clear to me. As I understand it, for a single replicate at one temperature, five petri dishes, each with ~200 eggs, was set up. Eight larvae were sampled "until the larvae were in the wandering stage." This procedure was repeated five times.

Does each point on Figure 5 represent $8 \times 5 = 40$ larvae? Please specify the sample size in the caption.

More detail is needed. If 8 larvae were sampled at each of the ages shown in Figure 5 during each of five replicates, that should be stated, as well as the total number of larvae measured for each age by temperature combination. Was one of the five petri dishes randomly selected at each age? Looking at 19 C, >70 ages were sampled, implying that > 500 of the original ~1000 (if no mortality, the proportion removed was probably greater) larvae were removed before the final sample. This could have considerably altered the growing conditions of the larvae, and should be discussed.

-The format of table 1 repeats a common mistake in the forensic entomology literature. In most of science, if an author publishes a mean and standard deviation, the study involved several measurements of that quantity. But in this or any other development study, the duration of a larval instar was not observed for any individual insect. The table deceives the reader. Furthermore, the format is not useful for the main purpose of this research, insect age estimation, because it does not show the proportion of each life stage observed as a function of age.

If one wants to use one variable to predict another, the relationship between the two must be specified. So to permit the use of instar to predict age, one needs to show the number of individuals in each life stage observed at each age. In other words, for each temperature there should be a table in which the rows are sample ages and the columns are life stages. The values should be counts of individuals observed to be that life stage.

Based on my interactions with other forensic entomologists, I anticipate some resistance to deleting Table 1 (as should be done). I will be satisfied if instead of doing that the authors present their data in this format as a supplementary file. This would be much closer to the raw data, and therefore more appropriate, than what is currently in the supplementary file.

MINOR COMMENTS

-The statement “the oldest progeny of Calliphorids on the corpse are often used to obtain a minimum postmortem interval (PMI_{min})” is misleading. In practice it is the oldest individual that was collected that is used to estimate PMI_{min}. Whether or not the set of insects collected from the corpse includes the earliest individual deposited as an egg or larva on the corpse is difficult to know.

-Given that no randomization method was specified, how is it that “Adults were randomly selected from [sic] each cage” and “eight larvae were randomly selected from the dish?” If the sample was not truly random, it may have been biased. (see International journal of Legal Medicine 129:405-410).

-What is the purpose of comparing *C. pinguis* to *C. megacephala*? It does not improve the interpretation of these results.

Review form: Reviewer 2

Is the manuscript scientifically sound in its present form?

Yes

Are the interpretations and conclusions justified by the results?

Yes

Is the language acceptable?

Yes

Is it clear how to access all supporting data?

Yes

Do you have any ethical concerns with this paper?

No

Have you any concerns about statistical analyses in this paper?

No

Recommendation?

Accept with minor revision (please list in comments)

Comments to the Author(s)

This manuscript investigates forensically-relevant development data from the blow fly *Chrysomya pinguis*, which is an important colonizer of human corpses and animal carcasses throughout Asia. It also represents the first developmental dataset for this species, which will no doubt be vital to forensic casework in the countries where this fly is found. My main concerns were focused on the establishment of a pure colony for experiments as well as the experimental procedure itself. More details in this regard would be very helpful.

Major comments:

Lines 66 – 77: This method of identification is slightly worrisome. Why weren't the adult flies that were originally collected in Suzhou definitively identified before establishing the laboratory colony? If there was even a single gravid female that wasn't *C. pinguis* (perhaps *C. megacephala*)

used as G0, the laboratory colony would be mixed and the results unreliable. These flies could have been missed during the random identification of adults emerged from eggs of the wild flies. Please add a justification as to why this identification route was taken as well as how you know for certain that you started with a pure colony.

Lines 79 – 83: The colony was maintained for over a year before new genetic material was introduced - why? Were any of the experiments performed BEFORE the May 2017 addition to the colony? This would make a big difference in interpreting the results as the colony would be quite inbred, which could affect the rate at which larvae developed. At the very least this deserves some attention in the discussion. Also, please indicate the generation of the colony for each experiment (e.g. G3 for the first round of experiments, G10 for the second round, etc.) in the methods.

Lines 88 – 91: If 1000 eggs divided into 5 petri dishes = 1 replicate, how many replicates were used per temperature per round of the experiment? How did you determine that you had 1000 eggs and what did you use to divide/separate the eggs? Was this one large cluster of eggs made from numerous females ovipositing in the same place, or many separate clutches of eggs dispersed on the pork?

There have been several studies in recent years discussing the relevance of population origin in developmental studies used in forensic entomology casework. Specifically, distinct geographic populations of conspecific blow flies can have significantly different developmental rates. It would be good to see a brief discussion of this phenomenon in the manuscript, potentially in a “future directions” aspect of this work.

Minor comments:

Line 37: “decomposition process of the ecosystem” reads strangely; change to “decomposition process of animals in most terrestrial ecosystems”

Line 49: change “Asia” to “Asian countries”

Line 55: change “regard” to “regarded”

Line 74: Remove “of”

Line 75: Change “form” to “from”

Line 86: How long were females protein-fed before inducing oviposition?

Line 96: The order here is confusing. Talk about the eggs and hatching before talking about the larvae in the previous paragraph.

Lines 97 – 99: Were eggs randomly assigned to each dish in the first place? How do you know that you didn't place a clutch of eggs from a single female in its own dish rather than a random sample of eggs from the initial 1000 eggs?

Line 104: Was each round of experiments performed for all temperatures simultaneously?

Lines 140: Paragraphs should be > 3 sentences. I would just combine this section into one paragraph.

Decision letter (RSOS-190003.R0)

19-Jun-2019

Dear Dr Wang,

The editors assigned to your paper ("Temperature-dependent development of the blow fly *Chrysomya pinguis* and its significance in estimating postmortem interval") have now received

comments from reviewers. We would like you to revise your paper in accordance with the referee and Associate Editor suggestions which can be found below (not including confidential reports to the Editor). Please note this decision does not guarantee eventual acceptance.

Please submit a copy of your revised paper before 12-Jul-2019. Please note that the revision deadline will expire at 00.00am on this date. If we do not hear from you within this time then it will be assumed that the paper has been withdrawn. In exceptional circumstances, extensions may be possible if agreed with the Editorial Office in advance. We do not allow multiple rounds of revision so we urge you to make every effort to fully address all of the comments at this stage. If deemed necessary by the Editors, your manuscript will be sent back to one or more of the original reviewers for assessment. If the original reviewers are not available, we may invite new reviewers.

- Data accessibility

<http://datadryad.org/submit?journalID=RSOS&manu=RSOS-190003>

- Competing interests

- Authors' contributions

- Acknowledgements

- Funding statement

on behalf of Dr Richard Benton (Associate Editor) and Kevin Padian (Subject Editor)
openscience@royalsociety.org

Comments to Author:

Reviewers' Comments to Author:

Reviewer: 1

Comments to the Author(s)

Chrysomya pinguis is a widespread and common blow fly. Published development data for *C. pinguis* are not available and could be very useful to death investigators. The manuscript is generally well written, and it potentially merits publication if the authors can revise the methods and data presentation.

MAJOR COMMENTS

-The sampling scheme is not clear to me. As I understand it, for a single replicate at one

temperature, five petri dishes, each with ~200 eggs, was set up. Eight larvae were sampled “until the larvae were in the wandering stage.” This procedure was repeated five times.

Does each point on Figure 5 represent $8 \times 5 = 40$ larvae? Please specify the sample size in the caption.

More detail is needed. If 8 larvae were sampled at each of the ages shown in Figure 5 during each of five replicates, that should be stated, as well as the total number of larvae measured for each age by temperature combination. Was one of the five petri dishes randomly selected at each age? Looking at 19 C, >70 ages were sampled, implying that > 500 of the original ~1000 (if no mortality, the proportion removed was probably greater) larvae were removed before the final sample. This could have considerably altered the growing conditions of the larvae, and should be discussed.

-The format of table 1 repeats a common mistake in the forensic entomology literature. In most of science, if an author publishes a mean and standard deviation, the study involved several measurements of that quantity. But in this or any other development study, the duration of a larval instar was not observed for any individual insect. The table deceives the reader. Furthermore, the format is not useful for the main purpose of this research, insect age estimation, because it does not show the proportion of each life stage observed as a function of age.

If one wants to use one variable to predict another, the relationship between the two must be specified. So to permit the use of instar to predict age, one needs to show the number of individuals in each life stage observed at each age. In other words, for each temperature there should be a table in which the rows are sample ages and the columns are life stages. The values should be counts of individuals observed to be that life stage.

Based on my interactions with other forensic entomologists, I anticipate some resistance to deleting Table 1 (as should be done). I will be satisfied if instead of doing that the authors present their data in this format as a supplementary file. This would be much closer to the raw data, and therefore more appropriate, than what is currently in the supplementary file.

MINOR COMMENTS

-The statement “the oldest progeny of Calliphorids on the corpse are often used to obtain a minimum postmortem interval (PMI_{min})” is misleading. In practice it is the oldest individual that was collected that is used to estimate PMI_{min}. Whether or not the set of insects collected from the corpse includes the earliest individual deposited as an egg or larva on the corpse is difficult to know.

-Given that no randomization method was specified, how is it that “Adults were randomly selected from [sic] each cage” and “eight larvae were randomly selected from the dish?” If the sample was not truly random, it may have been biased. (see International journal of Legal Medicine 129:405-410).

-What is the purpose of comparing *C. pinguis* to *C. megacephala*? It does not improve the interpretation of these results.

Reviewer: 2

Comments to the Author(s)

This manuscript investigates forensically-relevant development data from the blow fly *Chrysomya pinguis*, which is an important colonizer of human corpses and animal carcasses throughout Asia. It also represents the first developmental dataset for this species, which will no doubt be vital to forensic casework in the countries where this fly is found. My main concerns were focused on the establishment of a pure colony for experiments as well as the experimental procedure itself. More details in this regard would be very helpful.

Major comments:

Lines 66 – 77: This method of identification is slightly worrisome. Why weren't the adult flies that were originally collected in Suzhou definitively identified before establishing the laboratory colony? If there was even a single gravid female that wasn't *C. pinguis* (perhaps *C. megacephala*) used as G0, the laboratory colony would be mixed and the results unreliable. These flies could have been missed during the random identification of adults emerged from eggs of the wild flies. Please add a justification as to why this identification route was taken as well as how you know for certain that you started with a pure colony.

Lines 79 – 83: The colony was maintained for over a year before new genetic material was introduced - why? Were any of the experiments performed BEFORE the May 2017 addition to the colony? This would make a big difference in interpreting the results as the colony would be quite inbred, which could affect the rate at which larvae developed. At the very least this deserves some attention in the discussion. Also, please indicate the generation of the colony for each experiment (e.g. G3 for the first round of experiments, G10 for the second round, etc.) in the methods.

Lines 88 – 91: If 1000 eggs divided into 5 petri dishes = 1 replicate, how many replicates were used per temperature per round of the experiment? How did you determine that you had 1000 eggs and what did you use to divide/separate the eggs? Was this one large cluster of eggs made from numerous females ovipositing in the same place, or many separate clutches of eggs dispersed on the pork?

There have been several studies in recent years discussing the relevance of population origin in developmental studies used in forensic entomology casework. Specifically, distinct geographic populations of conspecific blow flies can have significantly different developmental rates. It would be good to see a brief discussion of this phenomenon in the manuscript, potentially in a "future directions" aspect of this work.

Minor comments:

Line 37: "decomposition process of the ecosystem" reads strangely; change to "decomposition process of animals in most terrestrial ecosystems"

Line 49: change "Asia" to "Asian countries"

Line 55: change "regard" to "regarded"

Line 74: Remove "of"

Line 75: Change "form" to "from"

Line 86: How long were females protein-fed before inducing oviposition?

Line 96: The order here is confusing. Talk about the eggs and hatching before talking about the larvae in the previous paragraph.

Lines 97 – 99: Were eggs randomly assigned to each dish in the first place? How do you know that you didn't place a clutch of eggs from a single female in its own dish rather than a random sample of eggs from the initial 1000 eggs?

Line 104: Was each round of experiments performed for all temperatures simultaneously?

Lines 140: Paragraphs should be > 3 sentences. I would just combine this section into one paragraph.

Author's Response to Decision Letter for (RSOS-190003.R0)

See Appendix A.

RSOS-190003.R1 (Revision)

Review form: Reviewer 1

Is the manuscript scientifically sound in its present form?

Yes

Are the interpretations and conclusions justified by the results?

Yes

Is the language acceptable?

Yes

Do you have any ethical concerns with this paper?

No

Have you any concerns about statistical analyses in this paper?

No

Recommendation?

Accept with minor revision (please list in comments)

Comments to the Author(s)

The authors have adequately addressed my concerns with one exception. They claim to have sampled larvae randomly from a rearing container. When reviewing the original manuscript, I questioned this claim "Given that no randomization method was specified." In the revised manuscript (line 98), the authors again describe sampling as random, but they do not describe any randomization procedure. It seems that sampling of larvae was not random, and because of the profound importance of an unbiased sample in so many aspects of science, the authors should not make this statement.

It is not possible for a human to reach into a petri dish and randomly sample maggots. An explicit randomization procedure would be required, which in this experiment would entail something like removing all larvae from the dish, giving each a computer-generated random number, and selecting the maggots with the 8 highest numbers. The unsampled larvae would be too disturbed to be used for older sample ages.

Decision letter (RSOS-190003.R1)

29-Jul-2019

Dear Dr Wang:

On behalf of the Editors, I am pleased to inform you that your Manuscript RSOS-190003.R1 entitled "Temperature-dependent development of the blow fly *Chrysomya pinguis* and its significance in estimating postmortem interval" has been accepted for publication in Royal Society Open Science subject to minor revision in accordance with the referee suggestions. Please find the referees' comments at the end of this email.

The reviewers and Subject Editor have recommended publication, but also suggest some minor revisions to your manuscript. Therefore, I invite you to respond to the comments and revise your manuscript.

- Ethics statement

- Data accessibility

If you wish to submit your supporting data or code to Dryad (<http://datadryad.org/>), or modify your current submission to dryad, please use the following link:
<http://datadryad.org/submit?journalID=RSOS&manu=RSOS-190003.R1>

- Competing interests

- Authors' contributions

- Acknowledgements

- Funding statement

Because the schedule for publication is very tight, it is a condition of publication that you submit the revised version of your manuscript before 07-Aug-2019. Please note that the revision deadline will expire at 00.00am on this date. If you do not think you will be able to meet this date please let me know immediately.

on behalf of Dr Richard Benton (Associate Editor) and Kevin Padian (Subject Editor)
openscience@royalsociety.org

Reviewer comments to Author:
Reviewer: 1

Comments to the Author(s)

The authors have adequately addressed my concerns with one exception. They claim to have sampled larvae randomly from a rearing container. When reviewing the original manuscript, I questioned this claim "Given that no randomization method was specified." In the revised manuscript (line 98), the authors again describe sampling as random, but they do not describe any randomization procedure. It seems that sampling of larvae was not random, and because of the profound importance of an unbiased sample in so many aspects of science, the authors should not make this statement.

It is not possible for a human to reach into a petri dish and randomly sample maggots. An explicit randomization procedure would be required, which in this experiment would entail something like removing all larvae from the dish, giving each a computer-generated random number, and selecting the maggots with the 8 highest numbers. The unsampled larvae would be too disturbed to be used for older sample ages.

Author's Response to Decision Letter for (RSOS-190003.R1)

See Appendix B.

Decision letter (RSOS-190003.R2)

09-Aug-2019

Dear Dr Wang,

I am pleased to inform you that your manuscript entitled "Temperature-dependent development of the blow fly *Chrysomya pinguis* and its significance in estimating postmortem interval" is now accepted for publication in Royal Society Open Science.

on behalf of Dr Richard Benton (Associate Editor) and Kevin Padian (Subject Editor)
openscience@royalsociety.org

Appendix A

Dear Reviewers,

Thank you very much for your careful review and constructive suggestions with regard to our manuscript. The comments are very valuable for us to revise and improve our paper. We have studied comments carefully and have made correction which we hope meet with approval. Revised portion are marked in red in the paper. Our responses to the comments are as following:

#Reviewer: 1

Comments to the Author(s):

Chrysomya pinguis is a widespread and common blow fly. Published development data for *C. pinguis* are not available and could be very useful to death investigators. The manuscript is generally well written, and it potentially merits publication if the authors can revise the methods and data presentation.

Response: Thanks very much for your time on revising our manuscript. We have revised the manuscript according to the recommendation.

MAJOR COMMENTS:

1. The sampling scheme is not clear to me. As I understand it, for a single replicate at one temperature, five petri dishes, each with ~200 eggs, was set up. Eight larvae were sampled “until the larvae were in the wandering stage.” This procedure was repeated five times.

Does each point on Figure 5 represent $8 \times 5 = 40$ larvae? Please specify the sample size in the caption.

In our study, for a single replicate at one temperature, five petri dishes, each dishes with ~200 eggs, was set up. Eight larvae were sampled from 1000 larvae every 4 hours until the larvae were in the wandering stage. This procedure was repeated five times. The reason why we separated 1000 eggs into 5 portions is because we sampled every 4 hours, so a large numbers of larvae are needed for sampling. In order to reduce the larval mass effect caused by the over-crowded larvae, the 1000 eggs were separated into 5 portions. We have clarified this in the caption of Figure 5, and the revisions are as follows: Curves of larval body length of *Chrysomya pinguis* variation with time at different temperatures. The vertical bars represent the standard deviation. Each point represents the average length of $8 \times 5 = 40$ larvae.

2. More detail is needed. 1) If 8 larvae were sampled at each of the ages shown in Figure 5 during each

of five replicates, that should be stated, as well as the total number of larvae measured for each age by temperature combination. 2) Was one of the five petri dishes randomly selected at each age? 3) Looking at 19 °C, >70 ages were sampled, implying that > 500 of the original ~1000 (if no mortality, the proportion removed was probably greater) larvae were removed before the final sample. This could have considerably altered the growing conditions of the larvae, and should be discussed.

Response: Thanks for your advice.

(1) We have specified the sample size in the caption. The details are as follows: Curves of larval body length of *Chrysomya pinguis* variation with time at different temperatures. The vertical bars represent the standard deviation. Each point represents the average length of $8 \times 5 = 40$ larvae.

(2) Yes, the dish was randomly selected. The dishes were numbered (marked 1, 2, 3, 4, 5). Before sampling, a series of random numbers were generated by a random number generator. The random numbers will determine which dish will be used for sampling.

(3) We have added a discussion regarding the effect sampling methods on the development, and the details are as follows: In forensic entomology development study, in order to record the body length and/or instar of a species, their larvae were usually reared at different temperatures, and individuals were removed periodically (usually without replacement) [38-41]. There are usually two sampling strategies. One is to select a narrower sampling interval at higher temperatures and a wider sampling interval at lower temperatures. The other is to select fixed sampling interval, just like the sampling method we used in the study. Though the method we used is more convenient as we don't need to change our sampling interval at different temperature, it usually consumes more samples, especially at lower temperature. For example, over half of the larvae (> 500) were sampled from the origin cohort at 16°C in this study, which may have a potential effect on the development of *C. pinguis*. An evaluation of sampling methods study by Wells et al. [42] revealed that random subsamples would produce a predictive model similar to that of the full data set, while subsampling the largest larvae produced a predictive model that performed poorly. The larvae were randomly sampled in this study, which means that our sampling method may have little effect on the development. Further studies are needed to verify this.

3. The format of table 1 repeats a common mistake in the forensic entomology literature. In most of

science, if an author publishes a mean and standard deviation, the study involved several measurements of that quantity. But in this or any other development study, the duration of a larval instar was not observed for any individual insect. The table deceives the reader.

Furthermore, the format is not useful for the main purpose of this research, insect age estimation, because it does not show the proportion of each life stage observed as a function of age.

If one wants to use one variable to predict another, the relationship between the two must be specified. So to permit the use of instar to predict age, one needs to show the number of individuals in each life stage observed at each age. In other words, for each temperature there should be a table in which the rows are sample ages and the columns are life stages. The values should be counts of individuals observed to be that life stage.

Based on my interactions with other forensic entomologists, I anticipate some resistance to deleting Table 1 (as should be done). I will be satisfied if instead of doing that the authors present their data in this format as a supplementary file. This would be much closer to the raw data, and therefore more appropriate, than what is currently in the supplementary file.

Response: Thank you very much for your suggestion. We have removed Table 1 from main text, and a supplementary file was added according to the recommendation.

MINOR COMMENTS:

4. The statement “the oldest progeny of Calliphorids on the corpse are often used to obtain a minimum postmortem interval (PMI_{min})” is misleading. In practice it is the oldest individual that was collected that is used to estimate PMI_{min}. Whether or not the set of insects collected from the corpse includes the earliest individual deposited as an egg or larva on the corpse is difficult to know.

Response: Thanks for your suggestion. The revisions are as follows: As they are often the first to arrive and oviposit on a dead body, the oldest immature individual of Calliphorids collected on the corpse are often used to obtain a minimum postmortem interval (PMI_{min}) for solving cases involving death.

5. Given that no randomization method was specified, how is it that “Adults were randomly selected from [sic] each cage” and “eight larvae were randomly selected from the dish?” If the sample was not truly random, it may have been biased. (see International journal of Legal Medicine 129:405-410).

Response: For larvae sampling, many authors collected subsamples that were clearly biased in that they targeted the largest individuals in the rearing container. In our study, eight larvae were randomly sampled from larval mass, not considering their size or stage.

For adults sampling, the target adults were male. So the ‘randomly’ used is not appropriate. We have deleted this word in the revised manuscript.

6. What is the purpose of comparing *C. pinguis* to *C. megacephala*? It does not improve the interpretation of these results.

Response: Thanks for your suggestion. We have deleted this confusing section.

#Reviewer: 2

Comments to the Author(s):

This manuscript investigates forensically-relevant development data from the blow fly *Chrysomya pinguis*, which is an important colonizer of human corpses and animal carcasses throughout Asia. It also represents the first developmental dataset for this species, which will no doubt be vital to forensic casework in the countries where this fly is found. My main concerns were focused on the establishment of a pure colony for experiments as well as the experimental procedure itself. More details in this regard would be very helpful.

Response: Thanks for your time on our manuscript. We have further clarified our colony establish methods in the following response.

Major comments:

1. Lines 66 – 77: This method of identification is slightly worrisome. Please add a justification as to why this identification route was taken as well as how you know for certain that you started with a pure colony. 1) Why weren't the adult flies that were originally collected in Suzhou definitively identified before establishing the laboratory colony? 2) If there was even a single gravid female that wasn't *C. pinguis* (perhaps *C. megacephala*) used as G0, the laboratory colony would be mixed and the results unreliable. These flies could have been missed during the random identification of adults emerged from eggs of the wild flies.

Response: We have confidence that the colony is pure, because (i) though adult *C. pinguis* has a very similar morphology with *C. megacephala*, they were easy to distinguish as *C. pinguis* has brilliant blue body color and gray bucca with black hairs. Eight female adults collected in Suzhou were identified as *C. pinguis* were used to establish colony. The identification conducted subsequently was used to revalidate using the male genitalia as the most reliable identification is based on the male genitalia differences between species. During the identification, all adults are identified as *C. pinguis*. (ii) The posterior spiracles of larval *C. megacephala* and *C. pinguis* are different. The distance between posterior spiracles in *C. megacephala* were much narrower than *C. pinguis* (Fig. 1). During the instar determination, all larvae examined are *C. pinguis*.

Fig. 1 A: *C. megacephala*; B: *C. pinguis*

2. Lines 79 – 83: The colony was maintained for over a year before new genetic material was introduced - why? Were any of the experiments performed BEFORE the May 2017 addition to the colony? This would make a big difference in interpreting the results as the colony would be quite inbred, which could affect the rate at which larvae developed. At the very least this deserves some attention in the discussion. Also, please indicate the generation of the colony for each experiment (e.g. G3 for the first round of experiments, G10 for the second round, etc.) in the methods.

Response: The development study of forensically important species is a major research direction of our laboratory. Several species are bred simultaneously in our laboratory. Only when the population reaches a certain size (for blow flies > 4000) the developmental experiments would be carried out. The number of eggs produced by the females is relatively small at the beginning of

study. After a long-term exploration, we found if we used a plastic bottle with pork to induce eggs, the numbers of eggs production will be increased. By this time, the Chinese Spring Festival is coming, so we reared the larvae at 19 °C to maintain the colony. In the second year, larvae were introduced to prevent the effect of inbred on development. After adults reached at a density of approximately 4000 adults, the study was carried out.

3. Lines 88 – 91: 1) If 1000 eggs divided into 5 petri dishes = 1 replicate, how many replicates were used per temperature per round of the experiment? 2) How did you determine that you had 1000 eggs and what did you use to divide/separate the eggs? 3) Was this one large cluster of eggs made from numerous females ovipositing in the same place, or many separate clutches of eggs dispersed on the pork?

Response: 1) In the experiment, in order to prevent larval-mass effect, we divided 1000 eggs collected in 1 hours into five groups and placed them in five petri dishes, and then put the five petri dishes into the same breeding box as a repetition. The eggs collected in 1 hours were only used in one repetition of one temperature, with a total of 7 temperatures, and each temperature was repeated 5 times.

2) The egg masses were weighed by an electronic balance with a precision of 0.0001 g, but it is only a rough estimation. The egg masses were divided by banister brush under stereoscopic microscope.

3) Two plastic bottles of 10 cm in height and 8 cm in diameter containing fresh lean pork were placed in the insect-rearing cage to induce oviposition. Eggs were collected in 1h from many separate clutches of eggs dispersed on the pork in the two plastic bottles above. Then, approximately 1000 eggs were estimated and divided by banister brush from these eggs and used as a replicate.

4. There have been several studies in recent years discussing the relevance of population origin in developmental studies used in forensic entomology casework. Specifically, distinct geographic populations of conspecific blow flies can have significantly different developmental rates. It would be good to see a brief discussion of this phenomenon in the manuscript, potentially in a “future directions” aspect of this work.

Response: Thanks very much for your suggestion. We have added a discussion regarding this aspect. The details are as follows: Recent studies have shown that geographically distinct populations can vary in developmental time [5, 39, 43]. Development time differences have been found in *Calliphora vicina* Robineau-Desvoidy, 1830 and *Lucilia sericata* (Meigen, 1826) (Diptera, Calliphoridae) from geographically separate populations [44-46]. For example, at 33.5°C, *L. sericata* from the Michigan strain developed fastest, whereas *L. sericata* from the California strain took the longest to mature. Differences between the findings for the same species could be a result of geographically separate populations. Therefore, it is important to collect precise development data in different geographic regions of the world so as to provide developmental data of the species in or around the particular crime scene when estimating the PMI_{min} [5, 47, 48].

Minor comments:

5. Line 37: “decomposition process of the ecosystem” reads strangely; change to “decomposition process of animals in most terrestrial ecosystems”

Response: Done.

6. Line 49: change “Asia” to “Asian countries”

Response: Done.

7. Line 55: change “regard” to “regarded”

Response: Done.

8. Line 74: Remove “of”

Response: Done.

9. Line 75: Change “form” to “from”

Response: Done.

10. Line 86: How long were females protein-fed before inducing oviposition?

Response: For this species, there are about a week that the females can complete ovary development and reproduction.

11. Line 96: The order here is confusing. Talk about the eggs and hatching before talking about the larvae in the previous paragraph.

Response: Thanks for your suggestion. The revisions are as follows:

The culture dishes were checked every hour to monitor the hatching of eggs. Once larvae began to hatch, meat slices about 5mm thick were placed in the petri dish to distribute larvae more evenly within the petri dish based on the larvae's food consumption. And one of the five dishes was chosen randomly in an order determined using a random number generator, and eight larvae were randomly selected from the dish every 4 hours until larvae were in the wandering stage. Samples were killed in hot water ($\geq 90^{\circ}\text{C}$) for 30 s and stored in 80% ethanol [33]. Samples were generally measured within a week of sampling. The larval length was measured by a Digital Vernier caliper with a precision of 0.01 mm (Shengong, Shanghai). The larval instar was determined using a calibrated stereomicroscope (Carl Zeiss, Göttingen, Germany). In the wandering and intra-pupal stages, observations were conducted every 4 h until the first adult emerged. The above experiment was repeated five times for each temperature in different incubators.

12. Lines 97 – 99: 1) Were eggs randomly assigned to each dish in the first place? 2) How do you know that you didn't place a clutch of eggs from a single female in its own dish rather than a random sample of eggs from the initial 1000 eggs?

Response: 1) The egg masses were weighed by an electronic balance with a precision of 0.0001 g, but it is only a rough estimation. The egg masses randomly were divided by banister brush under stereoscopic microscope.

2) It is hard to know whether a clutch of eggs from a single female or not. But during the experiment, the egg masses were mixed before dividing them into different dishes.

13. Line 104: Was each round of experiments performed for all temperatures simultaneously?

Response: Not all experiments performed simultaneously. The experimental progress is depended on the numbers of eggs the colony supplied.

14. Lines 140: Paragraphs should be > 3 sentences. I would just combine this section into one paragraph.

Response: Corrected. Thanks for your suggestion.

Appendix B

Reviewer comments to Author:

#Reviewer: 1

Comments to the Author(s)

1. The authors have adequately addressed my concerns with one exception. They claim to have sampled larvae randomly from a rearing container. When reviewing the original manuscript, I questioned this claim "Given that no randomization method was specified." In the revised manuscript (line 98), the authors again describe sampling as random, but they do not describe any randomization procedure. It seems that sampling of larvae was not random, and because of the profound importance of an unbiased sample in so many aspects of science, the authors should not make this statement.

It is not possible for a human to reach into a petri dish and randomly sample maggots. An explicit randomization procedure would be required, which in this experiment would entail something like removing all larvae from the dish, giving each a computer-generated random number, and selecting the maggots with the 8 highest numbers. The unsampled larvae would be too disturbed to be used for older sample ages.

Response: Thanks very much for your comments on how a randomization sample method should be conducted. We realized that our sampling method is not a ‘true’ randomization procedure, and the word ‘randomly’ has been deleted to avoid confusion according to the recommendation. Again, thank you very much for your work on revising our manuscript.